# Prompt-Based Editing for Text Style Transfer

**Guoqing Luo[†], Yu Tong Han[†], Lili Mou[†‡], Mauajama Firdaus[†]**

[†]Dept. Computing Science, Alberta Machine Intelligence Institute (Amii)
University of Alberta, Canada
[‡]Canada CIFAR AI Chair, Amii
{gluo, yhan22}@ualberta.ca
{doublepower.mou, mauzama.03}@gmail.com

## Abstract

Prompting approaches have been recently explored in text style transfer, where a textual prompt is used to query a pretrained language model (PLM) to generate style-transferred texts word by word in an autoregressive manner. However, such a generation process is less controllable and early prediction errors may affect future word predictions. In this paper, we propose a prompt-based editing approach to text style transfer. Specifically, we prompt a PLM for style classification and use the classification probability to compute a style score. Then, we perform discrete search with word-level editing to maximize a comprehensive scoring function for the style-transfer task. In this way, we transform a prompt-based generation problem into a classification one, which does not suffer from the error accumulation problem and is more controllable than the autoregressive generation of sentences. In our experiments, we performed both automatic and human evaluation on three style-transfer benchmark datasets, and show that our approach largely outperforms the existing systems that have 20 times more parameters. Additional empirical analyses further demonstrate the effectiveness of our approach.[1]

## 1 Introduction

Text style transfer aims to automatically rewrite a sentence by changing it from one style to another (John et al., 2019), such as transferring the positive-sentiment sentence "*He loves eating sandwiches*" into a negative one "*He hates eating sandwiches*". During the transfer, the style of the sentence must be changed, whereas the style-independent content should be preserved. Text style transfer has a wide range of real-world applications, such as personalized response generation (Yang et al., 2017; Zheng et al., 2021), text debiasing (Xiang et al., 2012; Ma et al., 2020),

text simplification (Dong et al., 2019; Kumar et al., 2020), and headline generation (Jin et al., 2020; Zhan et al., 2022).

Early work on text style transfer falls mainly into three categories: 1) Parallel supervision with labeled source–target sentence pairs in a sequence-to-sequence manner (Zhu et al., 2010; Rao and Tetreault, 2018; Zhang et al., 2020), 2) Non-parallel supervision with style labels only, including learning latent representations of style and content separately (Shen et al., 2017; John et al., 2019; Goyal et al., 2021) and constructing pseudo-parallel training data for learning (Luo et al., 2019; Krishna et al., 2020; Reid and Zhong, 2021), and 3) Unsupervised learning methods that do not require style labels (Jain et al., 2019; Xu et al., 2020).

Very recently, prompting methods have been explored in text style transfer (Reif et al., 2022; Suzgun et al., 2022), as large-scale pretrained language models (PLMs) enable us to perform various natural language generation tasks in a zero-shot (Wei et al., 2022a; Sanh et al., 2022) or exemplar-based manner (Brown et al., 2020; Schick and Schütze, 2021a). In this paper, we also follow the prompt-based setting. This does not require any training samples or labels, but directly performs inference with PLMs; thus, it is more challenging than the above three settings.

In previous work, a prompt (e.g., a piece of text "Rewrite the text to be positive:") is used to query a PLM, which will then generate a style-transferred sentence in an autoregressive manner (Reif et al., 2022; Suzgun et al., 2022). However, such autoregressive generation is less controllable as words are generated one after another by the PLM. It has the error accumulation problem where early errors of the PLM will affect its future predictions, leading to less satisfactory performance in general.

To this end, we propose a prompt-based editing approach to unsupervised style transfer. We first design a PLM-based style scorer. Specifically,

---

[1]Our code and resources are available at: https://github.com/MANGA-UOFA/Prompt-Edit

we prompt a PLM for style classification and use the classification probability to compute a style score. Then, we perform steepest-ascent hill climbing (SAHC; Russell and Norvig, 2010) for discrete search with word-level editing (such as replacement, insertion, and deletion) to maximize a heuristically defined scoring function for style transfer. In this way, we transform a prompt-based generation problem into a classification one, which involves only a style-word prediction and is generally believed to be easier than multiple-word predictions for sentence generation.

Our approach provides several additional advantages. First, it does not suffer from the error accumulation problem, because it performs word edits scattered throughout the entire sentence instead of generating a sentence word by word. Further, we design a discrete search algorithm that combines the PLM-based style score with other scoring functions, including fluency and semantic similarity. Consequently, the search algorithm contributes to a more controllable and refined generation of sentences.

We used Eleuther AI's GPT-J-6B (an off-the-shelf PLM)[2] and conducted both automatic and human evaluations on three style-transfer benchmark datasets. Results show that our prompt-based editing approach largely outperforms existing prompting systems that have 20 times more parameters. Additional empirical analysis shows the effectiveness of different scoring components and the search algorithm proposed in our approach.

## 2   Related Work

**Prompting.** Prompting methods use a piece of text to query a PLM to provide desired outputs (Liu et al., 2021). The simplest prompting method, perhaps, is zero-shot prompting (Wei et al., 2022a; Sanh et al., 2022; Suzgun et al., 2022), which directly queries a PLM to perform a natural language processing task, but this may result in less well-formatted or logical sentences (Reif et al., 2022). Another method is few-shot prompting (Brown et al., 2020; Schick and Schütze, 2021a,b; Wei et al., 2022b). It requires several task-specific examples for PLM, but can achieve higher performance than zero shot prompting, and is thus more widely adopted in NLP tasks (Schick and Schütze, 2021a; Brown et al., 2020; Wei et al., 2022b).

Prompting methods were initially applied to

[2]https://github.com/kingoflolz/mesh-transformer-jax

natural language classification tasks (Schick and Schütze, 2021a,b; Min et al., 2022), where PLMs are asked to predict the masked word given a piece of text containing the token "[MASK]", and the predicted word is then projected to a label by a predefined verbalizer. With the emergence of various PLMs (Devlin et al., 2019; Radford et al., 2019; Brown et al., 2020; Raffel et al., 2020; Wei et al., 2022a), prompting methods have recently been widely applied to natural language generation tasks (Liu et al., 2021), such as text style transfer (Reif et al., 2022; Suzgun et al., 2022) and machine translation (Radford et al., 2019; Brown et al., 2020; Raffel et al., 2020).

**Text style transfer.** Traditionally, style-transfer generation can be accomplished by supervised methods with parallel training data (Xu et al., 2012; Zhang et al., 2015; Rao and Tetreault, 2018). However, obtaining parallel data is labor-intensive and time-consuming, which remains a significant challenge for this task.

To mitigate the need for parallel data, one line of research is non-parallel supervision, where it trains the model on a non-parallel but style-labeled corpus (Shen et al., 2017). Early work focuses on learning latent representations of content and style separately (Hu et al., 2017; Fu et al., 2018; John et al., 2019; Bao et al., 2019). Goyal et al. (2021) train multiple language models as discriminators for each of the target styles given the content representation. However, explicit separation of content and style is not always possible, because style can only be conveyed holistically for some sentences.

On the other hand, researchers construct pseudo-parallel training data for training the model in a supervised manner (Lample et al., 2018; Li et al., 2018; Luo et al., 2019; Krishna et al., 2020). Reid and Zhong (2021) first train an attentive style classifier to synthesize source–target style pairs, which are then used to train a Levenshtein editor and perform multi-span edits. However, the process of constructing pseudo-parallel data can sometimes yield poor-quality data, which would lead to sub-optimal training and low model performance.

Another line of research is devoted to unsupervised learning methods, where the training samples contain no style labels (Jain et al., 2019; Xu et al., 2020). Jain et al. (2019) train an encoder–decoder model with unlabeled texts and multiple controlling scorers to perform formal style transformation. However, these unsupervised learning methods re-

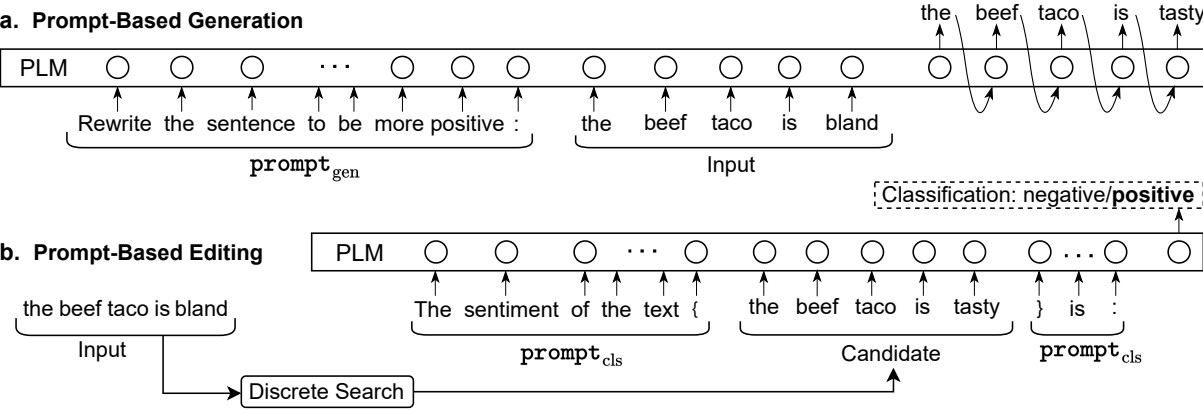

Figure 1: a) Prompt-based generation: previous work (Reif et al., 2022) uses a prompt to query a PLM, which generates a style-transferred sentence in an autoregressive manner. b) Our prompt-based editing approach involves one-word classification (e.g., *positive* or *negative* in sentiment transfer).

quire a complicated training process, which is not efficient.

Recently, researchers have developed several prompt-based approaches that generate style-transferred texts in a zero-shot (Suzgun et al., 2022) or exemplar-based manner (Reif et al., 2022). Such methods do not require a learning process or any training labels. Reif et al. (2022) prompt large-scale PLMs to generate sentences in various styles. Suzgun et al. (2022) generate multiple candidate sentences and then use a re-ranking mechanism to choose one with the highest score as the final output.

Our approach follows the prompt-based setting and directly performs style transfer without any training procedure. However, unlike other work that mainly performs autoregressive generation, our approach proposes a new prompt-based editing paradigm for text generation, where we not only design a PLM-based scoring function but also develop a discrete search algorithm that is particularly suited to our scenario.

## 3 Approach

Given an input sentence $\mathbf{x} = (x_1, \cdots, x_m)$, our goal is to generate a sentence $\mathbf{y} = (y_1, \cdots, y_n)$ that transfers the style of $x$. Figure 1b depicts the framework of our prompt-based editing approach, where we propose to prompt a pretrained language model (PLM) to predict the style of a candidate sentence. Then, we perform discrete search and iteratively edit the candidate sentence to maximize a scoring objective that involves the PLM's classification probability. Finally, the highest-scored candidate is taken as the style-transferred sentence.

### 3.1 Prompt-Based Classifier

In previous work, researchers directly prompt a PLM to obtain style-transferred sentences (Figure 1a; Reif et al., 2022; Suzgun et al., 2022). However, this could be a challenging process, as the PLM has to generate the sentence in a zero-shot or exemplar-based manner; such a process is autoregressive and less controllable.

To address this, we design a prompt-based classifier, transforming style-transfer text generation into a classification problem. Specifically, we prompt a PLM for a style word, which in turn yields a style score. This involves only one single-step prediction and is much simpler than generating the entire sentence.

Given a candidate sentence $[\mathbf{y}]$, we intuitively design the prompt as

$$\mathtt{prompt}_{\mathrm{cls}}(\mathbf{y}) \equiv \text{The [t] of the text \{ [\mathbf{y}] \} is:} \quad (1)$$

where [t] is the style-transfer task, i.e., *sentiment* or *formality* in our experiments, and "{" and "}" are text boundary markers (Reif et al., 2022). Notice that we have not performed prompt engineering, which is beyond the scope of this paper. Instead, our focus is to develop a prompt-based editing approach for text style transfer.

Based on the above prompt, we perform the next-word prediction to obtain a style probability. Specifically, the PLM computes the conditional probability of the next word w in the vocabulary given the prompt, denoted by $P_{\mathrm{PLM}}(\mathrm{w} \,|\, \mathtt{prompt}_{\mathrm{cls}}(\mathbf{y}))$.

We denote $s_i$ by the representative word of the $i$th style. Here, $s_i$ is simply chosen to be the most intuitive style word, namely,

*positive* and *negative* for sentiment transfer and *formal* and *informal* for formality transfer. In general, the predicted probabilities of the two styles are $P_{\mathrm{PLM}}(\mathrm{s}_1 \,|\, \texttt{prompt}_{\mathrm{cls}}(\mathbf{y}))$ and $P_{\mathrm{PLM}}(\mathrm{s}_2 \,|\, \texttt{prompt}_{\mathrm{cls}}(\mathbf{y}))$.

To compute the style score, we consider the ratio of the two styles. Suppose a sentence in style $\mathrm{s}_1$ is to be transferred to $\mathrm{s}_2$, we design the style score as:

$$f_{\mathrm{sty}}(\mathbf{y}) = \frac{P_{\mathrm{PLM}}(\mathrm{s}_2 \,|\, \texttt{prompt}_{\mathrm{cls}}(\mathbf{y}))}{P_{\mathrm{PLM}}(\mathrm{s}_1 \,|\, \texttt{prompt}_{\mathrm{cls}}(\mathbf{y}))} \qquad (2)$$

Such a ratio measures the candidate's relative affiliation with different styles.[3] It is more robust than the predicted target-style probability $P_{\mathrm{PLM}}(\mathrm{s}_2|\texttt{prompt}_{\mathrm{cls}}(\mathbf{y}))$, which could be affected by the data sample *per se*.

## 3.2 Search Objective

We apply an edit-based search for unsupervised style transfer. This follows the recent development of search-based text generation (Li et al., 2020; Kumar et al., 2020; Liu et al., 2022; Dong et al., 2021; Jolly et al., 2022), where local edits (e.g., word changes) are performed to maximize a heuristically defined objective function. However, different from previous search-based work, we propose to prompt an off-the-shelf PLM to compute a style score and do not require any task-specific training procedure. Overall, our objective function involves three aspects:

$$f(\mathbf{y}; \mathbf{x}) = f_{\mathrm{sty}}(\mathbf{y}) \cdot f_{\mathrm{flu}}(\mathbf{y}) \cdot f_{\mathrm{sem}}(\mathbf{y}, \mathbf{x}) \quad (3)$$

where the style scorer $f_{\mathrm{sty}}$ is designed in Section 3.1; $f_{\mathrm{flu}}$ and $f_{\mathrm{sem}}$ are fluency and semantic similarity scorers, mostly adopted from previous work and explained below.

**Language Fluency.** A language model scorer provides an approximation of how fluent a candidate sentence $\mathbf{y}$ is. We follow Suzgun et al. (2022) and use GPT-2 (Radford et al., 2019) to obtain the fluency score of the candidate $\mathbf{y}$ by the geometric mean of predicted probabilities:

$$f_{\mathrm{flu}}(\mathbf{y}) = \left( \left[ \prod_{i=1}^{t} P_{\mathrm{GPT\text{-}2}}(y_i|\mathbf{y}_{<i}) \right]^{\frac{1}{t}} \right)^{\alpha} \quad (4)$$

---

**Algorithm 1** Prompt-Based Editing

1: **Input**: Original sentence $\mathbf{x}$, iterative steps $T$
2: $\mathbf{y}^{(0)} = \mathbf{x}$
3: **for** $t \in \{1, \cdots, T\}$ **do**
4:      Enumerate all edit positions and operations
5:      Obtain the highest-scored candidate $\mathbf{y}^*$ by Eqn. (3)
6:      **if** $f_{\mathrm{sty}}(\mathbf{y}^*) > 1$    ▷ *PLM believes style transferred*
7:        **then: return** $\mathbf{y}^*$
8:      **if** $f(\mathbf{y}^{(t-1)}, \mathbf{x}) \geq f(\mathbf{y}^*, \mathbf{x})$   ▷ *Local optimum found*
9:        **then: return** $\mathbf{y}^{(t-1)}$
10:      **else:** $\mathbf{y}^{(t)} = \mathbf{y}^*$
11: **return** $\mathbf{y}^{(T)}$

---

where $\alpha$ is a hyperparameter balancing $f_{\mathrm{flu}}$ with other scoring functions[4].

**Semantic Similarity.** The semantic similarity scorer evaluates how an output $\mathbf{y}$ captures the semantics of an input $\mathbf{x}$. In our work, we adopt word- and sentence-level semantic similarities as in Li et al. (2020).

A word-level scorer focuses on keyword information, where the keywords in the input sentence $\mathbf{x}$ are extracted by the Rake system (Rose et al., 2010). Then, the RoBERTa model (Liu et al., 2019) is adopted to compute the contextualized representation, denoted by $\mathrm{RBT}(\mathrm{w}, \mathbf{s})$, for a word $\mathrm{w}$ in some sentence $\mathbf{s}$. The word-level semantic score is defined as the lowest similarity among all the keywords, given by

$$f_{\mathrm{word}}(\mathbf{y}, \mathbf{x}) = \min_{\mathrm{k} \in \mathrm{keyword}(\mathbf{x})} \max_{y_i \in \mathbf{y}} \cos(\mathrm{RBT}(\mathrm{k}, \mathbf{x}), \mathrm{RBT}(y_i, \mathbf{y}))$$

$$(5)$$

A sentence-level scorer computes the cosine similarity of two sentence vectors as $f_{\mathrm{sent}}(\mathbf{y}, \mathbf{x}) = \cos(\boldsymbol{y}, \boldsymbol{x}) = \frac{\boldsymbol{y}^{\top}\boldsymbol{x}}{||\boldsymbol{y}|| \cdot ||\boldsymbol{x}||}$, where the sentence vectors $\boldsymbol{y}$ and $\boldsymbol{x}$ are also encoded by RoBERTa.

Finally, the semantic similarity score is computed as the product of word- and sentence-level scores:

$$f_{\mathrm{sem}}(\mathbf{y}, \mathbf{x}) = f_{\mathrm{word}}(\mathbf{y}, \mathbf{x})^{\beta} \cdot f_{\mathrm{sent}}(\mathbf{y}, \mathbf{x})^{\gamma} \quad (6)$$

where $\beta$ and $\gamma$ are the weighting hyperparameters.

## 3.3 Discrete Search Algorithm

We perform style-transfer generation by discrete local search using editing operations, such as word insertion, deletion, and replacement, following previous work (Miao et al., 2019; Li et al., 2020). However, we propose to use steepest-ascent hill

---

[3]While our datasets only consider the transfer between two styles, our approach can be extended to multiple styles in a one-vs-one or one-vs-all manner.

[4]Notice that a weighting hyperparameter is not needed for the style scorer $f_{\mathrm{sty}}$ because the relative weight of different scorers are given in $f_{\mathrm{flu}}$ and $f_{\mathrm{sem}}$.

climbing (SAHC; Russell and Norvig, 2010) as our search algorithm.

During development, we measured the edit distance between the input sentences and the reference outputs for sentiment and formality transfer tasks. Our observation is that the average edit distance is 2.9 steps for sentiment transfer and 4.7 steps for formality transfer. Therefore, we set the maximum number of edit steps to 5 to maintain their resemblance. This, unfortunately, makes previous search algorithms—such as simulated annealing (SA; Liu et al., 2020) and first-choice hill climbing (FCHC; Schumann et al., 2020)—ineffective, as they cannot fully make use of the limited search steps.

In our work, we use the SAHC algorithm: in a search step $t$, SAHC enumerates every editing position and performs every editing operation (namely, word deletion, replacement, and insertion)[5]. Then it selects the highest-scored candidate sentence $\mathbf{y}^*$ if the score $f(\mathbf{y}^*, \mathbf{x})$ is higher than $f(\mathbf{y}^{(t-1)}, \mathbf{x})$ before it reaches the maximum edit steps. Otherwise, SAHC terminates and takes the candidate $\mathbf{y}^{(t-1)}$ as the style-transferred output. In this way, our SAHC greedily finds the best edit for every search step and is more powerful than SA and FCHC in our scenario, as will be shown in Section 4.6.

Moreover, we design an additional stopping criterion such that the search terminates when the prompted PLM predicts that the source style has changed into the target one even if it has not reached the maximum edit steps. This not only improves time efficiency but also encourages content preservation.

Our approach is summarized in Algorithm 1.

# 4 Experiments

In this section, we will present empirical evaluation of our proposed prompt-based editing approach. First, we will introduce our datasets and setups. Then, we will show our main results, followed by detailed analyses.

## 4.1 Datasets

We evaluated our approach on two standard style-transfer tasks: sentiment and formality.

We used Yelp reviews (YELP; Zhang et al., 2015) and Amazon reviews (AMAZON; He and McAuley, 2016) for sentiment transfer. These two datasets

have been widely used in previous work (Luo et al., 2019; John et al., 2019; Suzgun et al., 2022). YELP contains reviews for restaurants and other businesses (Zhang et al., 2015), and AMAZON contains product reviews. Both the YELP and AMAZON datasets contain 500 positive and 500 negative sentences in the test set.

For formality transfer, we used Grammarly's Yahoo Answers Formality Corpus (GYAFC; Rao and Tetreault, 2018). GYAFC consists of sentences that were extracted from the Yahoo Answers forum. We chose the Family & Relationships domain following Luo et al. (2019) and Suzgun et al. (2022). The test set contains 500 formal and 500 informal sentences.

## 4.2 Implementation Details

We used Eleuther AI's off-the-shelf GPT-J-6B as the prompt-based classifier for computing the style score. We also used the off-the-shelf pretrained language model RoBERTa-Large (Liu et al., 2019) to encode sentences (Section 3.2) and to predict top-$k$ words as candidate edits (Section 3.3). We set $k = 50$ for all sentiment and formality transfer datasets.

For the weighting hyperparameters $\alpha$, $\beta$, and $\gamma$ of the search objective $f(\mathbf{y})$ in Eqn. (3), they were $\frac{1}{4}$, $\frac{1}{6}$, and $\frac{1}{6}$ for both YELP and AMAZON datasets, and $\frac{1}{4}$, $\frac{3}{8}$, and $\frac{3}{8}$ for the GYAFC dataset. This shows that the style scorer is the most important one among all the scorers.

We developed our prompt-based editing approach with Python 3.7 and Pytorch 1.11.0. The experiments were conducted on NVIDIA A100 SXM4 GPUs.

## 4.3 Evaluation Metrics

We adopted the following automatic evaluation metrics:

- **Style transfer accuracy.** This measures whether a generated output is correctly transferred. Following the practice in Reif et al. (2022) and Lai et al. (2021), we used SiEBERT (Hartmann et al., 2022) for sentiment classification, and finetuned a RoBERTa-Large (Liu et al., 2019) for formality classification.
- **BLEU.** The BLEU score measures the semantic similarity between generated outputs and human-written references. Following Luo et al. (2019) and Reif et al. (2022), we used `multi-bleu.perl` to obtain the BLEU-4 score.

---

[5]For replacement and insertion, we follow Li et al. (2020) and choose top-$k$ candidate words predicted by RoBERTa due to efficiency concerns.

| Setting | Method | Model | #Para (B) | YELP | | | | AMAZON | | | |
|---|---|---|---|---|---|---|---|---|---|---|---|
| | | | | ACC% | BLEU | GM | HM | ACC% | BLEU | GM | HM |
| Zero-shot | Vanilla | LLM | 128 | 69.7* | 28.6* | 44.6 | 40.6 | - | - | - | - |
| | | LLM-dialog | 128 | 59.1* | 17.6* | 32.3 | 27.1 | - | - | - | - |
| | P&R† | GPT-J-6B | 6 | 68.6 | 19.8 | 35.2 | 30.1 | 57.1 | 21.7 | 35.2 | 31.4 |
| | Ours | GPT-J-6B | 6 | **73.0** | **40.1** | **54.1** | **51.7** | 72.7 | 28.6 | 45.6 | 41.0 |
| Few-shot | Distant exemplars | GPT-J-6B | 6 | 52.8 | 35.8 | 43.5 | 42.7 | 51.0 | 27.1 | 37.2 | 35.4 |
| | | GPT-3 curie | 6.7 | 53.0* | 48.3* | 50.6 | 50.5 | 72.2 | 22.9 | 40.7 | 34.8 |
| | | LLM | 128 | 79.6* | 16.1* | 35.8 | 26.8 | - | - | - | - |
| | | LLM-dialog | 128 | **90.6*** | 10.4* | 30.7 | 18.7 | - | - | - | - |
| | | GPT-3 danvinci | 175 | 74.1* | 43.8* | 57.0 | 55.1 | 87.3 | 28.3 | 49.7 | 42.7 |
| | P&R† | GPT-J-6B | 6 | 75.0 | 42.5 | 56.5 | 54.3 | 66.8 | 20.5 | 37.0 | 31.4 |
| | Ours | GPT-J-6B | 6 | 74.5 | **48.9** | **60.3** | **59.0** | 78.5 | **37.1** | **54.0** | **50.4** |

Table 1: Results on YELP and AMAZON test sets. #Para: Number of parameters. GM and HM: Geometric mean and harmonic mean of ACC% and BLEU. †We replicated Prompt & Rerank (Suzgun et al., 2022) by their released code, as the settings in Suzgun et al. (2022) are incompatible with other previous work. *Quoted from Reif et al. (2022). Other results are given by our experiments. The performance of LLM and LLM-dialog is not available for AMAZON because these PLMs are not public.

| Method | Model | #Para (B) | ACC% | BLEU | GM | HM |
|---|---|---|---|---|---|---|
| Distant exemplars | GPT-J-6B | 6 | 39.4 | 33.1 | 36.1 | 36.0 |
| P&R | GPT-J-6B | 6 | **44.4** | 32.9 | 38.2 | 37.8 |
| Ours | GPT-J-6B | 6 | **44.4** | **33.4** | **38.5** | **38.1** |

Table 2: Four-shot performance on the GYAFC dataset, considering both directions of informal ↔ formal.

- **Geometric mean (GM)** and **harmonic mean (HM)**. They are the average of the above-mentioned metrics, evaluating the overall performance of text style transfer. Again, this follows the standard practice in previous work (Luo et al., 2019; Li et al., 2020).

We also performed human evaluation on selected style-transfer systems, detailed in Section 4.6.

### 4.4 Baselines

Since our approach is based on prompting and does not require a training process, we compare our approach with the following existing prompting systems:

- **Vanilla prompting.** This baseline method prompts a PLM with "Here is some text: { [x] }. Here is a rewrite of the text, which is more [s]: {" where [x] is the input and [s] is the style word. This baseline directly obtains a style-transferred sentence, shown in Figure 1a. No exemplars are used here.
- **Distant-exemplar prompting.** We adopted the approach in Reif et al. (2022), which queries a large PLM (such as the LLM, LLM-dialog, and 175B-parameter GPT-3[6]) with several style-

transfer exemplars in a few-shot manner. However, their exemplars have different target styles from the test cases in the inference task, and thus we call it *distant-exemplar prompting*.

- **Prompt & Rerank.** Suzgun et al. (2022) propose a method that generates multiple candidate outputs from different manually designed prompts; then, they rerank the outputs by a heuristically defined scoring function. It should be mentioned that the paper (Suzgun et al., 2022) adopts a setting that is non-compatible with other work: they report different directions of sentiment transfer separately, while excluding informal-to-formal transfer in the formality experiment. Therefore, we replicated their work under the standard settings (Luo et al., 2019; Reif et al., 2022).

To the best of our knowledge, Reif et al. (2022) and Suzgun et al. (2022) are the only prior studies of prompting methods on text style transfer.

### 4.5 Main Results

Table 1 shows the performance of different prompting systems on YELP and AMAZON datasets. Com-

---

[6]We use the same prompt provided by Reif et al. (2022) to obtain results on the off-the-shelf GPT-3 babbage and GPT-J-6B for the YELP, AMAZON, and GYAFC datasets.

| Dataset | Method | Style | Content | Fluency | Average |
|---------|--------|-------|---------|---------|---------|
| YELP | Prompt & Rerank | 3.64 | 3.55 | 3.04 | 3.41 |
| | Our approach | **3.76** | **4.24** | **3.13** | **3.71** |
| AMAZON | Prompt & Rerank | 3.46 | 3.52 | 3.38 | 3.45 |
| | Our approach | **3.67** | **3.97** | **3.62** | **3.75** |

Table 3: Human evaluation on the sentiment transfer datasets. We show human ratings of style transfer strength (Style), content preservation (Content), and fluency.

| Dataset | Model | ACC% | BLEU | GM | HM | PPL |
|---------|-------|------|------|----|----|-----|
| YELP | Full model | 73.0 | **40.1** | 54.1 | 51.7 | 122.7 |
| | w/o style | 17.9 | 25.1 | 21.2 | 33.9 | 29.3 |
| | w/o semantic | 74.0 | 39.0 | 53.7 | 51.1 | 124.0 |
| | w/o fluency | **81.3** | 39.3 | **56.5** | **53.0** | 223.6 |
| | w/o stop criterion | 78.3 | 25.2 | 44.4 | 38.1 | 192.4 |
| AMAZON | Full model | 72.7 | **28.6** | 45.6 | 41.0 | 137.2 |
| | w/o style | 33.6 | 20.2 | 26.1 | 25.3 | 31.5 |
| | w/o semantic | 71.1 | 28.1 | 44.7 | 40.3 | 116.3 |
| | w/o fluency | 78.0 | **28.6** | **47.2** | **41.8** | 229.9 |
| | w/o stop criterion | **79.9** | 19.3 | 39.3 | 31.1 | 176.3 |

Table 4: Ablation study on the sentiment transfer datasets in the zero-shot setting. PPL: Perplexity (the smaller, the better). In the "w/o style" setting, the model mainly optimizes toward $f_{flu}$, so it achieves an extraordinarily low PPL; however, its style is usually not transferred, shown by extraordinarily low ACC%. Therefore, this is not a meaningful style-transfer setting and is grayed out.

| Dataset | Algorithm | ACC% | BLEU | GM | HM |
|---------|-----------|------|------|----|----|
| YELP | SAHC | **73.0** | **40.1** | **54.1** | **51.7** |
| | FCHC | 67.2 | 31.8 | 46.2 | 43.1 |
| | SA | 66.0 | 28.7 | 43.5 | 40.0 |
| AMAZON | SAHC | **72.7** | **28.6** | **45.6** | **41.0** |
| | FCHC | 64.1 | 24.8 | 39.8 | 35.7 |
| | SA | 63.2 | 23.7 | 38.7 | 34.4 |

Table 5: Results of different search algorithms on the sentiment transfer datasets.

pared with the recently proposed Prompt & Rerank system (Suzgun et al., 2022), our approach achieves a performance improvement of 14 and 3 points for GM, as well as 15 and 5 points for HM in the zero- and few-shot settings, respectively, averaged across the two datasets. Further, compared with 175B-parameter GPT-3 with distant exemplars (i.e., style-transfer exemplars containing source texts and outputs written in non-target styles), our approach yields higher GM and HM by more than 3 and 5 points, respectively, also averaged across the two datasets. This is a compelling result, as our approach yields a better balance between content preservation and style transfer strength while using a 20x smaller PLM.

Table 2 shows the results of different prompting systems on the GYAFC dataset, where both informal-to-formal and formal-to-informal directions are considered (Luo et al., 2019; Reif et al., 2022). For a fair comparison with previous prompting systems, we followed Suzgun et al. (2022) and conducted experiments in a four-shot setting. As seen, our method outperforms previous approaches in GM and HM scores, which is consistent with the results in Table 1. It is also noticed that our approach achieves less improvement on GYAFC than on YELP and AMAZON, as formality transfer is more challenging than sentiment transfer.

## 4.6 Detailed Analyses

In this subsection, we conduct in-depth analyses to assess the effectiveness of our prompt-based editing approach. Due to the limit of time and resources, we chose the sentiment transfer datasets (YELP and AMAZON) as our testbed.

**Human Evaluation.** We conducted human evaluation via pairwise comparison of system outputs to further confirm the superiority of our approach. Specifically, we randomly selected 100 outputs from the recently proposed Prompt-and-Rerank (P&R) system (Suzgun et al., 2022) and

our approach is based on the same GPT-J-6B model. Following Luo et al. (2019) and Krishna et al. (2020), we asked three human annotators, who were instructed to rate each sentence based on a 1–5 Likert scale (Stent et al., 2005) in terms of style transfer strength, content preservation, and fluency (Briakou et al., 2021). Our annotations were strictly blind; the samples from the two prompting approaches were randomly shuffled and the annotators did not know which approach generated the sample.

We measured the inter-rater agreement by Fleiss' Kappa score (1971) for the Likert scale ratings. They are 0.37, 0.42, and 0.39 for style transfer strength, content preservation, and fluency, respectively. These scores are considered fair correlation[7].

Table 3 presents the results of human evaluation. We observe that our prompt-based editing approach outperforms P&R in all three aspects, particularly in terms of content preservation. This is because with the proposed stopping criterion and discrete

---

[7]https://en.wikipedia.org/wiki/Fleiss%27_kappa

| YELP | Negative ⟶ Positive | Positive ⟶ Negative |
|---|---|---|
| Source | so far i'm not really impressed | their lunch special is a great value |
| P&R | *The text is good now* | but their lunch is a *great* value |
| Ours | so far i'm really impressed | their lunch special is not a great value |
| AMAZON | Negative ⟶ Positive | Positive ⟶ Negative |
| Source | i like neutrogena products as a rule, so this was a disappointment. | for my purpose this is the perfect item. |
| P&R | i like neutrogena products, so this was a *disappointment*. | for my purpose this is the *perfect* item. *So this text has two different purposes: to be a text and to be a rewrite...* |
| Ours | overall i like neutrogena products as a rule, so this was a success. | but for my purpose this is not the perfect item. |
| GYAFC | Informal ⟶ Formal | Formal ⟶ Informal |
| Source | think about what good it brought about. | i'm unsure concerning what i should do. |
| P&R | think about what good it *will bring about* ... | i'm *not certain* about what to do *next...* |
| Ours | please think about what all the good news has brought about. | yeah lol really ... i'm unsure concerning what i 'll do. |

Table 6: Example outputs on the YELP, AMAZON, and GYAFC datasets. Improperly generated expressions are italicized.

search, we avoid unnecessary edits and preserve the original content well. Our approach also achieves a higher average score, which is consistent with the automatic evaluation results in Table 1, further demonstrating the effectiveness of our approach.

**Ablation Study.** To evaluate the contribution of key components in our model, we conducted an ablation study of different scoring functions and our proposed stopping criterion.

Table 4 shows that all the scorers play a role in our approach, and that the prompt-based style scorer is the most important one. This makes sense, as it is the only signal of the style, without which we would not be able to perform meaningful style transfer. Moreover, we find that the fluency scorer slightly hurts style accuracy and BLEU scores, which are the standard metrics in Luo et al. (2019). However, it significantly improves language model probability (i.e., lower perplexity), which roughly estimates the fluency of text (John et al., 2019). Therefore, we deem the fluency scorer $f_{flu}$ essential to our text style transfer model.

In addition, our approach involves a stopping criterion that terminates the search process if the PLM believes the style is successfully transferred. As seen from the last row of Table 4, more edit steps (w/o stop criterion) improve the style accuracy but drastically hurt BLEU scores. This shows that our stopping criterion is able to seek a balance between style transfer accuracy and content preservation.

**Discrete Search Algorithms.** Our steepest-ascent hill climbing (SAHC) algorithm enumerates candidate edits, including word deletion, insertion, and replacement (where top-50 candidate words are considered for efficiency concerns). Then, SAHC selects the best one for the next round of editing, shown in Algorithm 1.

We compare our SAHC with two stochastic optimization algorithms, first-choice hill climbing (FCHC; Schumann et al., 2020) and simulated annealing (SA; Liu et al., 2020), which are used in previous search-based text generation. Both FCHC and SA perform stochastic local changes to obtain a candidate sentence. If the proposed sentence is better than the current one, the algorithms will accept the new candidate. Otherwise, FCHC will keep the current candidate, while SA may still accept the candidate with a small probability.

From Table 5, we observe that our SAHC algorithm significantly outperforms FCHC and SA in both style-transfer accuracy and the BLEU score. This is likely due to the limited number of edit steps, requiring that the algorithm should make an effective edit at every search step. The results confirm that SAHC is more suited in our scenario than other discrete search algorithms

**Case Study.** We show in Table 6 that our method is able to avoid issues that arise from the error accumulation problem in an autoregressive generation. This is observed through several example outputs by P&R and our approach for YELP, AMAZON, and GYAFC datasets. We see that the previous approach, which performs autoregressive generation, yields less controllable and satisfactory sentences. For example, given the source input "for my purpose this is the perfect item" in the

positive-to-negative sentiment transfer of the AMA-ZON dataset, P&R generates an unrelated sentence starting with "So this text has", leading to the subsequent improper word predictions "a text and to be a rewrite".

However, our prompt-based editing approach transfers the sentiment of a source sentence from positive to negative by inserting the words *but* and *not*, while maintaining other semantic content. This shows that our approach is capable of generating more sensible and controllable sentences.

In addition, we find that our approach is able to convert the style of source input with multiple edits. For example, given the source sentence "i'm unsure concerning what i should do" in formal-to-informal transfer, our approach inserts multiple tokens (*yeah*, *lol*, *really*, and "...") at the beginning and replaces *should* with *'ll* at the end, and the sentence is transferred to an informal one. By allowing iterative edits and examining all possible positions and editing operations, we are able to have multiple word edits scattered throughout the sentence and experience a gradual transfer of style.

## 5 Conclusion

In this paper, we propose a novel prompt-based editing approach to text style transfer that turns a prompt-based generation problem into a classification one. It does not suffer from the issue of error accumulation and is more controllable than autoregressive generation. Our experiments on three benchmark datasets show that the proposed approach significantly outperforms the existing prompting systems while using 20x fewer parameters. Additional empirical analysis shows the effectiveness of different scorers and the discrete search algorithm in our approach.

## 6 Limitation

Our paper transforms a generation problem into a classification one in text style transfer, but it comes with a trade-off between output quality and inference efficiency. Nevertheless, our algorithm can be implemented in a highly parallel manner when evaluating different candidates, and we only need five iterations. Therefore, the efficiency of our SAHC is already much higher than other search algorithms (such as SA) which requires several hundred search steps (Liu et al., 2020). Further, the efficiency can be improved by learning from the search results (Li et al., 2020), i.e., fine-tuning a PLM based on our

search outputs. In this way, our approach can be more computationally efficient.

Another limitation is the need for manually designed prompts, which is inevitable in zero-shot prompting. Our current work adopts the most intuitive prompt and has not performed prompt engineering. In the future, we would like to investigate prompt tuning (Schick and Schütze, 2021b; Li and Liang, 2021; Wei et al., 2022a) to mitigate the reliance on designing prompts.

## Acknowledgments

We thank all reviewers and chairs for their valuable comments. This research is supported in part by the Natural Sciences and Engineering Research Council of Canada (NSERC) under Grant No. RGPIN-2020-04465, the Amii Fellow Program, the Canada CIFAR AI Chair Program, the Alberta Innovates Program, a UAHJIC project, a donation from Deep-Mind, and the Digital Research Alliance of Canada (alliancecan.ca).

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
