# OpenReview forum: "Prompt-Based Editing for Text Style Transfer"
_EMNLP/2023/Conference — EMNLP 2023 Findings_

### Official Review · Reviewer_bmvG · 2023-08-02

**Soundness:** 4

**Excitement:**

4: Strong: This paper deepens the understanding of some phenomenon or lowers the barriers to an existing research direction.

**Paper Topic And Main Contributions:**

This paper focuses on how to better utilize large langeuage models for text style transfer. The specific approach involves using prompt-based editing approach as alternatives to traditional generation methods.

The contributions are that this paper proposes a novel prompt-based editing approach to text style transfer that turns a prompt-based generation problem into a classification one. It does not suffer from the issue of error accumulation and is more controllable than the autoregressive generation. The experiments on three benchmark datasets show that the proposed approach significantly outperforms the state-of-the-art prompting systems.

**Reasons To Accept:**

1.The prompt-based editing method proposed in this paper can be applied to similar problems that involve generating outputs using large models, particularly in scenarios where the generated outputs need to mostly retain consistency with the input. It has a certain degree of universality.
2.The experimental results of this paper demonstrate excellent performance, and the analysis provided is also detailed and comprehensive.

**Reasons To Reject:**

 How to use the search method to generate candidate text is not very clear. It is recommended to provide a more detailed description.

**Reproducibility:**

4: Could mostly reproduce the results, but there may be some variation because of sample variance or minor variations in their interpretation of the protocol or method.

**Reviewer Confidence:**

4: Quite sure. I tried to check the important points carefully. It's unlikely, though conceivable, that I missed something that should affect my ratings.

---

> ### Author Rebuttal · Authors · 2023-08-28
>
> Thank you for reading our paper and providing valuable feedback. Additionally, thank you for saying our approach is “novel” and our experimental analysis is “detailed and comprehensive”!
>
> **Part 4, Q1**: “How to use the search method to generate candidate text is not very clear. It is recommended to provide a more detailed description.”
>
> We thank the reviewer for the valuable suggestion. Should the paper be accepted, we will make sure to explain in more detail about our search method in Section 3.3 and highlight it in Algorithm 1 as well.

---

### Official Review · Reviewer_xk33 · 2023-08-03

**Soundness:** 3

**Excitement:**

3: Ambivalent: It has merits (e.g., it reports state-of-the-art results, the idea is nice), but there are key weaknesses (e.g., it describes incremental work), and it can significantly benefit from another round of revision. However, I won't object to accepting it if my co-reviewers champion it.

**Paper Topic And Main Contributions:**

This paper proposes a combination of modules to do text style transfer. The authors apply an edit-based search method, the steepest-ascent hill climbing (SAHC),  to generate a series of candidate sentences by inserting, deleting, and replacing words of the original sentences. And they choose the best candidate by defining a scoring function with three aspects – style (using the GBT-J-6B model), fluency (using the GPT2 model), and semantic similarity (using the RoBERTa model). From the experiments, the proposed method outperforms three PLM-based baslines.

**Questions For The Authors:**

1.	Why do the authors choose the three factors – style, fluency, and semantic similarity? They are necessary. But are they enough?
2.	What will happen if the authors replace the PLM with another style classifier (eg., BERT-based models) even though prompt-based classifier is currently popular?


**Reasons To Accept:**

1.	The goal of this paper to generate a sentence with a different style is very clear and the path to solving the problem is also very clear.
2.	The idea of using the PLM as a style classifier is relatively new to some extent, even though it follows the same routine as the authors’ previous work in sentence simplification [1].
3.	The statement of the methodology is good. The implementation is comprehensive. The results are solid.

[1] Dhruv Kumar, Lili Mou, Lukasz Golab, and Olga Vechtomova. 2020. Iterative Edit-Based Unsupervised Sentence Simplification. In Proceedings of the 58th Annual Meeting of the Association for Computational Linguistics, pages 7918–7928.


**Reasons To Reject:**

1.	The work is old wine in a new bottle. The core is the SAHC algorithm to ‘enumerate every editing position and perform every editing operation’. We can see from Table 5 that it plays a very significant role. The PLMs and prompt-based editing here seem not so important.
2.	The completeness of the work cannot cover the weak flaws in the core content. This is essentially a sentence editing work rather than a sentence generation work. Sentence editing seems more controllable than generation. However, we can use it to do the sentiment and formality style transfer work. But it may be difficult to do a modern-to-Shakespeare style transfer task. I don’t mean this is not a good work. It just tends to be conservative, especially there’s already similar framework used in their previous work.
3.	The method of this paper is like the Prompt & Rerank [2] method, while the latter is more generation-based.

[2] Mirac Suzgun, Luke Melas-Kyriazi, and Dan Jurafsky. 2022. Prompt-and-Rerank: A Method for Zero-Shot and Few-Shot Arbitrary Textual Style Transfer with Small Language Models. In Proceedings of the 2022 Conference on Empirical Methods in Natural Language Processing, pages 2195–2222.


**Reproducibility:**

5: Could easily reproduce the results.

**Reviewer Confidence:**

5: Positive that my evaluation is correct. I read the paper very carefully and I am very familiar with related work.

---

> ### Author Rebuttal · Authors · 2023-08-28
>
> **General Response**:
>
> We thank the reviewer for the detailed feedback and questions, and for saying that “the statement of the methodology is good”!
>
> We surmise that a general theme that the reviewer’s questions revolve around is the novelty of our work, such as a similar editing framework used in [Kumar et al., 2020], the PLMs and prompt-based editing seem not so important in the approach, and the method is like the Prompt & Rerank method.
>
> We acknowledge that our work follows some practice in previous work, but still provides new insights into the prompt-based paradigm of text style transfer. We first design a prompt-based classifier for style classification, which is recognized by the reviewer. Further, we involve an appropriate modification of the search algorithm and use the steepest-ascent hill climbing variant as well as a stopping criterion to make the style transfer process more controllable. The search algorithm is not only very important in our approach, as shown in Table 5, but also a substantial novelty compared with previous prompting literature (e.g., Prompt & Rerank) that is mainly about prompt engineering. We sincerely hope that this difference is able to set us apart from other literature enough to make a meaningful contribution to this line of work.
>
> While the reviewer points out that the SAHC algorithm, which is one of the main contributions, plays a significant role in our prompt-based editing approach, it is concerned that the PLMs here seem not so important. In actuality, PLMs play an integral part in our approach along with SAHC. We use PLMs to measure different aspects of the search objectives. In particular, the prompt-based classifier also serves as a style scorer, being a novel insight into our work. The contributions of different PLM-based scorers are shown in Table 4, and detailed in “Ablation Study” in Section 4.5 (Lines 500–516). Therefore, we believe that both the SAHC algorithm and the PLMs are contributive enough to our prompt-based editing approach.
>
> We answered other questions individually.
>
>
> **Part 3, Point 2**: "The completeness of the work cannot cover the weak flaws in the core content...Sentence editing seems more controllable than generation. However, we can use it to do the sentiment and formality style transfer work. But it may be difficult to do a modern-to-Shakespeare style transfer task..."
>
> We want to point out that formality is a style that is not concretely defined in a sentence, which is as challenging as the Shakespeare-to-modern task. Nevertheless, we are conducting additional experiments on it. While it takes time to obtain quantitative results, we show some example outputs here:
>
> | Input | Output | Reference |
> |------|------|------|
> |Wisely and slow .| Do Wisely and also slowly . |Go wisely and slowly .|
> |What hast thou found ?|What have you found out ? |What have you found out ?|
> | And is he a man to encounter Tybalt ? | Is he man enough to see Tybalt ? | Is he man enough at this point to face off with Tybalt ?
>
> Overall, our approach has the potential to perform the Shakespeare-to-modern transfer task.
>
> **Part 4, Q1**: “Why do the authors choose the three factors – style, fluency, and semantic similarity? They are necessary. But are they enough?”
>
> We follow [1-2] and use these three factors to evaluate the quality of sentences comprehensively. We think they are enough as these factors are believed to be a common practice of evaluation [3], and we will clarify this point in the revision.
>
> [1] Mirac Suzgun, Luke Melas-Kyriazi, and Dan Jurafsky. 2022. Prompt-and-Rerank: A Method for Zero-Shot and Few-Shot Arbitrary Textual Style Transfer with Small Language Models. In EMNLP, pages 2195–2222.
>
> [2] Jingjing Li, Zichao Li, Lili Mou, Xin Jiang, Michael Lyu, and Irwin King. 2020. Unsupervised text generation by learning from search. In NeurIPS, pages 10820–10831.
>
> [3] Di Jin, Zhijing Jin, Zhiting Hu, Olga Vechtomova, and Rada Mihalcea. 2022. Deep Learning for Text Style Transfer: A Survey. Computational Linguistics, 48(1):155–205.
>
>
> **Part 4, Q2**: “What will happen if the authors replace the PLM with another style classifier (eg., BERT-based models) even though prompt-based classifier is currently popular?”
>
> In our approach, PLM-based search objectives are introduced to perform text style transfer without any training samples or labels (Lines 60–64), allowing us to achieve remarkable performance. However, if we replace the PLM with another style classifier, such as a BERT-based model, it would require a large labeled dataset to finetune the classifier and may achieve a better performance, but this does not constitute a fair comparison to previous prompt-based methods.
>
> We thank the reviewer again for the detailed comments, and will thoroughly revise our paper according to the suggestions provided. We hope our response could alleviate all the concerns and are looking forward to your stronger support!

---

### Official Review · Reviewer_PLiK · 2023-08-05

**Soundness:** 3

**Excitement:**

3: Ambivalent: It has merits (e.g., it reports state-of-the-art results, the idea is nice), but there are key weaknesses (e.g., it describes incremental work), and it can significantly benefit from another round of revision. However, I won't object to accepting it if my co-reviewers champion it.

**Missing References:**

None come to mind.

**Paper Topic And Main Contributions:**

This paper introduces a discrete token edit algorithm to generate a token at every timestep t using a lightweight PLM to construct a sentence of the target style. In this greedy approach, the best token edit is chosen w.r.t a style score predicted by the underlying PLM for that phrase at time t. Hence a typically generative process for style transfer is converted into a sort of greedy search without training the underlying model.

This work addresses the error accumulation problem commonly seen with simple prompt-based approaches. It experimentally seems to perform the best w.r.t its SOTA baselines.

The approach is simple but effective-seeming and a novel direction for performing token generation.

**Questions For The Authors:**

1. Please address my questions above regarding the accuracy of the method i.e. whether it will be feasible with more complex timestep variant styles (say author style or discourse)
2. Is this extendable to the setting where a style is not concretely defined? For example, if a style is represented as a few-shot scheme i.e. a group of sentences of one style, is it possible to then come up with an accurate style score?
3. Why wasn't a non-generative classifier considered to produce the style score? This would need training but it might have been able to perform better in a low-resource setting.
4. Is there any other work which uses an edit search algorithm in a generative process similar to this?

**Reasons To Accept:**

1. The idea is interesting and seems novel. It is an intuitive way of breaking down the style transfer problem and guiding the tokens appropriately at every time step t.
2. It shows to perform the best experimentally.
3. The main reason I think this might be helpful to the community is simply the idea of using a score to greedily determine the next token. This simple idea might be borrowed in future work to aid generation, not necessarily in the context of style transfer.

**Reasons To Reject:**

1. The approach has an inherent flaw in that manually tuning the mean edit distance is needed for every style/dataset. Therefore it might not be usable at all in the situation in a low resource setting and relies too much on fine-tuned hyperparameters. These max token thresholds are fundamentally problematic, especially in the case of a complex style which does not involve simple word edits and we can assume that the output length ~ input length. Perhaps there is a more elegant want to determine these rules and give the model more flexibility in sentence structure and length

2. This also in a way suffers from error accumulation in the same way that prompting-based approaches do. If the output at t is non-optimal, the algorithm from timestep t will deteriorate more with time suffering exactly the same problem as prompting-based approaches. The suggestion that the error accumulation problem is alleviated using examples in Table 6 is simply not valid evidence of this. More experiments are required to justify this claim.

3. It is only feasible for simple styles which rely on simple token edits. This approach might work well in a zero-shot setting for more complex styles as well. However this cannot be assumed, and this work only tests two very simple styles.

4. It is computationally more expensive and potentially a lot slower and practically infeasible. At every time step t there are 4 extra scoring prompts (one for each edit type) to generate the next token and this might dramatically increase inference time. I do not see any discussion about the feasibility of this in a practical setting or how it compares computationally to other models.

5. The underlying greedy assumption that any output at a given timestep t seems like an over assumption and can fail for more complex styles. It relies solely on the fact that the style score at timestep t is an accurate oracle in informing all future decisions. This might fail in the context of a style that contains long-term logical dependencies in a more bidirectional work. Style and formality are more time invariant in that sense and so this approach is more suited to this kind of setting.

6. The style considered i.e. sentiment and formality are outdated and too simple to justify SOTA performance in the style transfer task.

**Reproducibility:**

4: Could mostly reproduce the results, but there may be some variation because of sample variance or minor variations in their interpretation of the protocol or method.

**Reviewer Confidence:**

5: Positive that my evaluation is correct. I read the paper very carefully and I am very familiar with related work.

**Typos Grammar Style And Presentation Improvements:**

The writing in this paper can be substantially improved. Generally in the punctuation, grammer and style of writing too sometimes. Please consider going through all the text again and trimming now excess punctuation.

---

> ### Author Rebuttal · Authors · 2023-08-28
>
> **General Response**:
>
> We thank the reviewer for the detailed feedback and questions. We surmise that a general theme that the reviewer’s questions revolve around is: sentiment and formality transfer are too simple, will the approach be extended to more complex timestep variant styles (author style)?
>
> We acknowledge that sentiment transfer is relatively simple to perform, but formality is a style that is not concretely defined in a sentence, which is as challenging as the author style transfer task, i.e., Shakespeare-to-modern transfer. Nevertheless, we are conducting additional experiments on this task. While it takes time to obtain quantitative results, we show some example outputs here:
>
> | Input | Output | Reference |
> |------|------|------|
> |Wisely and slow .| Do Wisely and also slowly . |Go wisely and slowly .|
> |What hast thou found ?|What have you found out ? |What have you found out ?|
> | And is he a man to encounter Tybalt ? | Is he man enough to see Tybalt ? | Is he man enough at this point to face off with Tybalt ?
>
> Apart from formality transfer, we believe our prompt-based editing approach has the potential to perform other complex style transfer tasks, such as author style transfer.
>
> We answered other questions individually.
>
> **Part 3, Point 2**: “This also in a way suffers from error accumulation in the same way that prompting-based approaches do. If the output at t is non-optimal, the algorithm from timestep t will deteriorate more with time suffering exactly the same problem as prompting-based approaches. The suggestion that the error accumulation problem is alleviated using examples in Table 6 is simply not valid evidence of this. More experiments are required to justify this claim.”
>
> We acknowledge that our edit-based approach still occasionally encounters the error accumulation issue where non-optimal sentences are chosen for future search. However, our approach still greatly alleviates the issue compared with the prompt-based generation, where an LM performs the next word prediction simply based on its categorical prediction distribution without any explicit control over the quality of generated sentences. On the contrary, our approach applies search objectives for explicit control during the edits of sentences, and this has been proved to be effective in numerous previous works [Kumar et al., 2020; Li et al., 2020; Liu et al., 2020]
>
>
> **Part 3, Point 4**: “It is computationally more expensive and potentially a lot slower and practically infeasible. At every time step t there are 4 extra scoring prompts (one for each edit type) to generate the next token, and this might dramatically increase inference time. I do not see any discussion about the feasibility of this in a practical setting or how it compares computationally to other models.”
>
> First, we would like to point out that there is a misunderstanding. We only use one prompt for an off-the-shelf PLM to perform style classification (Line 225), and we do not use any other prompts to perform the edit types.
>
> Second, we mention in the limitation section that our approach can be implemented in a highly parallel manner when evaluating different candidates, and the search algorithm only needs a few iterations, so our approach is more efficient than some stochastic search algorithms requiring several hundred search steps. (Lines 602–612)
>
> **Part 4, Q2**: “Is this extendable to the setting where a style is not concretely defined? For example, if a style is represented as a few-shot scheme i.e. a group of sentences of one style, is it possible to then come up with an accurate style score?”
>
> Yes, it is extendable. In fact, formality is a style that is not concretely defined and is thus a challenging task. We mention in Lines 451–456 that we use four exemplars as a few-shot scheme. Specifically, we group two formal and two informal sentences together and feed them to an off-the-shelf PLM to help it understand what “informal” and “formal” mean. The results in Table 2 show that it is possible to give an accurate style score.
>
> **Part 4, Q3**: “Why wasn't a non-generative classifier considered to produce the style score? This would need training but it might have been able to perform better in a low-resource setting.”
>
> In our approach, PLM-based search objectives are introduced to perform text style transfer without any training samples or labels (Lines 60–64), allowing us to achieve remarkable performance. However, if we replace the PLM with another style classifier, such as a BERT-based model, it would require a large labeled dataset to finetune the classifier and may achieve a better performance, but this does not constitute a fair comparison to previous prompt-based methods.
>
> **Part 4, Q4**: “Is there any other work that uses an edit search algorithm in a generative process similar to this?”
>
> Yes. However, different from previous work in the edit-based generation, our approach designs a prompt-based classifier, which serves as a style scorer during discrete search and does not require any training procedure (Lines 262--265). Further, we involve an appropriate modification of the search algorithm and use the steepest-ascent hill climbing variant as well as a stopping criterion to make the style transfer process more controllable.
>
> **Part 5, Point 1** “The writing in this paper can be substantially improved. Generally in the punctuation, grammer and style of writing too sometimes. Please consider going through all the text again and trimming now excess punctuation.”
>
> We thank the reviewer for the suggestion. However, we would like to point out that this paper has been carefully proofread by multiple professionals and native speakers, and we could not easily find format issues or errors in grammar and punctuation. In addition, we wrote the paper following the guidelines in [1], and we did not use excess punctuation. If the reviewer could give any concrete examples of grammatical errors or excess punctuation, we would be happy to address them in the revision.
>
> [1] Strunk, William Jr., and E.B. White. The Elements of Style. 4th ed., Longman, 2000.

---

### Meta-Review · Area_Chair_nbFb · 2023-09-19

**Recommendation:** 3

**Metareview:**

The paper presents an editing mechanism which first prompts a PLM to extract style class of a given text, which is then used along with word-level editing for style transfer. The motivation is claimed to be a better more control during the style transfer phase, in contrast with conventional prompt-based style transfer solutions. On 3 style-transfer benchmarks, the paper outperforms previous SotA. The discussion with xk33 and PLiK clarified some of the missing steps and better contextualised the contribution of the work compared to previous sentence-editing literature. I encourage the authors to take these suggestions on board.

---

### Decision · Program_Chairs · 2023-10-07

**Decision:**

Accept-Findings

**Comment:**

The paper presents an editing mechanism which first prompts a PLM to extract style class of a given text, which is then used along with word-level editing for style transfer. The motivation is claimed to be a better more control during the style transfer phase, in contrast with conventional prompt-based style transfer solutions. On 3 style-transfer benchmarks, the paper outperforms previous SotA. The discussion with xk33 and PLiK clarified some of the missing steps and better contextualised the contribution of the work compared to previous sentence-editing literature. I encourage the authors to take these suggestions on board.